# Analysis of lncRNA Expression Profile during the Formation of Male Germ Cells in Chickens

**DOI:** 10.3390/ani10101850

**Published:** 2020-10-11

**Authors:** Wen Gao, Chen Zhang, Kai Jin, Yani Zhang, Qisheng Zuo, Bichun Li

**Affiliations:** 1Key Laboratory of Animal Breeding Reproduction and Molecular Design for Jiangsu Province, College of Animal Science and Technology, Yangzhou University, Yangzhou 225009, China; gaowen1104@126.com (W.G.); m160647@yzu.edu.cn (C.Z.); jinkai0621@163.com (K.J.); ynzhang@yzu.edu.cn (Y.Z.); 006664@yzu.edu.cn (Q.Z.); 2Joint International Research Laboratory of Agriculture and Agri-Product Safety of Ministry of Education of China, Yangzhou University, Yangzhou 225009, China

**Keywords:** chicken, lncRNAs, embryonic stem cell, primordial germ cell, spermatogonial stem cell

## Abstract

**Simple Summary:**

The differentiation of germ cells plays an important role in sex differentiation in poultry. Therefore, it is necessary for us to explore the potential regulators in the process of germ cell development. In this study, RNA-seq was used to detect the expression profile of long non-coding RNAs (lncRNAs) in chicken embryonic stem cells (ESCs), primordial germ cells (PGCs) and spermatogonial stem cells (SSCs). The results showed that a total of 296, 280 and 357 differentially expressed lncRNAs (DELs) were screened in ESCs vs. PGCs, ESCs vs. SSCs and PGCs vs. SSCs, respectively. Functional analysis of the target genes of DELs showed that autophagy, Wnt/β-catenin, TGF-β, Notch and ErbB signaling pathways were involved in the differentiation process of male germ cells and, moreover, XLOC_612026, XLOC_612029, XLOC_240662, XLOC_362463, XLOC_023952, XLOC_674549, XLOC_160716, ALDBGALG0000001810, ALDBGALG0000002986, XLOC_657380674549, XLOC_022100 and XLOC_657380 were predicted to be the key lncRNAs in this process. Our findings could not only supply scientific data for constructing the gene regulatory network of germ cell development, but also provide new ideas for further optimizing the induction efficiency of germ cells in vitro.

**Abstract:**

Germ cells have an irreplaceable role in transmitting genetic information from one generation to the next, and also play an important role in sex differentiation in poultry, while little is known about epigenetic factors that regulate germ cell differentiation. In this study, RNA-seq was used to detect the expression profiles of long non-coding RNAs (lncRNAs) during the differentiation of chicken embryonic stem cells (ESCs) into spermatogonial stem cells (SSCs). The results showed that a total of 296, 280 and 357 differentially expressed lncRNAs (DELs) were screened in ESCs vs. PGCs, ESCs vs. SSCs and PGCs vs. SSCs, respectively. Gene Ontology (GO) and KEGG enrichment analysis showed that DELs in the three cell groups were mainly enriched in autophagy, Wnt/β-catenin, TGF-β, Notch and ErbB and signaling pathways. The co-expression network of 37 candidate DELs and their target genes enriched in the biological function of germ cell development showed that XLOC_612026, XLOC_612029, XLOC_240662, XLOC_362463, XLOC_023952, XLOC_674549, XLOC_160716, ALDBGALG0000001810, ALDBGALG0000002986, XLOC_657380674549, XLOC_022100 and XLOC_657380 were the key lncRNAs in the process of male germ cell formation and, moreover, the function of these DELs may be related to the interaction of their target genes. Our findings preliminarily excavated the key lncRNAs and signaling pathways in the process of male chicken germ cell formation, which could be helpful to construct the gene regulatory network of germ cell development, and also provide new ideas for further optimizing the induction efficiency of germ cells in vitro.

## 1. Introduction

Germ cells are a kind of special cell in sexually reproducing organisms which play a pivotal role in transmitting genetic information from ancestors to future generations. Recently, inducing embryonic stem cells (ESCs) into germ cells, such as primordial germ cells (PGCs) and spermatogonial stem cells (SSCs), by adding cytokines and chemical induction reagents to culture medium or through transgenic technologies, has an become important way to explore the formation and differentiation mechanisms of germ cells [1,2,3]. These studies provided new strategies for the development of regenerative medicine and the treatment of reproductive diseases like infertility. However, the low efficiency of in vitro induction has always been an important factor limiting the transformation of this technology. Therefore, it is urgent to find key factors that influence germ cell development from a new perspective.

Long non-coding RNAs (lncRNAs) are a class of non-coding RNAs with transcripts > 200 nucleotides in length and had no potential for translation. Intriguingly, the mechanism of action by which lncRNAs regulate gene expression is generally acknowledged at three levels, which are epigenetic, transcriptional and post-transcriptional levels. For instance, lncRNAs can interface with chromatin remodeling machinery in several ways, including acting as signal lncRNAs or scaffold lncRNAs by recruiting chromatin remodeling complexes to specific gene loci [4,5]. In addition, lncRNAs can work as ceRNAs to modulate transcription by sequestering transcription factors, catalytic proteins or miRNAs [6].

To date, a large number of lncRNAs have been found to participate in regulating biological processes such as cell proliferation, apoptosis and autophagy [7,8]. Moreover, lncRNAs have great significance in regulating the fate of stem cells, such as maintaining cellular pluripotency [9]. By carrying out high-throughput sequencing analysis of RNA derived from the nucleus and cytoplasm of embryonic stem cells (ESCs), Guo et al. [10] found that there were significant differences in the processing and subcellular localization of lncRNAs between human and mouse ESCs, which led to different biological functions of lncRNAs in the regulation of cellular pluripotency. Recent studies have shown that lncRNAs also play a pivotal regulatory role in the differentiation and development of germ cells. As the progenitor of germ cells, the differentiation and development of PGCs are regulated by internal transcription factors and signals released by neighboring somatic cells [11,12]. However, there are still few studies on the differentiation of PGCs regulated by lncRNAs. In 2019, Zuo et al. [13] reported that transcription factor p53 and high levels of H3K4me2 could activate N6-methylated lncRNA PGC transcript-1 (lncPGCAT-1) to positively regulate the formation of chicken PGCs in vivo and vitro through the MAPK signaling pathway, which expanded the understanding of the function and molecular mechanism of lncRNAs in PGC development. Spermatogonial stem cells (SSCs), as a class of unique male reproductive stem cells, play an important role in spermatogenesis and maintaining male fertility. A large number of studies have shown that lncRNAs are the key regulators of SSCs. Li et al. [14] found a new spermatogonia-specific lncRNA (lncRNA033862), which can regulate the self-renewal and survival of mouse SSCs by regulating the expression of Gfra1. Weng et al. [15] discovered that 473 lncRNAs were specifically expressed in the testes of 60-day-old boars containing only undifferentiated SSCs by using RNA Sequencing, demonstrating that these lncRNAs may play a potential role in the self-renewal of porcine SSCs. Moreover, a few studies have shown that cross-talk between lncRNAs and other epigenetic factors is involved in spermatogenesis. Hu et al. [16] reported that lncRNA AK015322 promoted the proliferation of mouse spermatogonial germ cell line C18-4 by serving as a decoy of microRNA-19b-3p and attenuating the expression of its target transcriptional factor ETV5, which was a pivotal gene for SSC self-renewal. Meiotic recombination hot spot locus (mrhl) RNA, a long non-coding RNA, was revealed to negatively regulate Wnt signaling through its protein partner Ddx5/p68 in mouse spermatogonial cells [17].

Although there is increasing evidence that lncRNAs play a pivotal role in spermatogenesis, most studies on lncRNAs are mainly focused on humans and mice, and fewer on chickens. What is more, most of the studies that have been done in chickens are basically focused on identifying lncRNAs in specific tissues or specific developmental stages, while few studies are about how lncRNAs regulate the differentiation of chicken ESCs into SSCs. Herein, RNA-seq was employed to identify the expression profile of lncRNAs in chicken ESCs, PGCs and SSCs, and the potential functions and related signaling pathways of crucial lncRNAs in the formation of male germ cells were systematically excavated, which will not only help to construct the gene regulatory network during the development of male chicken germ cells, but also provide a theoretical basis for optimizing the induction system of the differentiation of ESCs into SSCs.

## 2. Materials and Methods

### 2.1. Ethics Statement

All experimental procedures in the present study were reviewed and approved by the Institutional Animal Care and Use Committee of Yangzhou University (approval number: 151-2014). Procedures were performed in accordance with the Regulations for the Administration of Affairs Concerning Experimental Animals (Yangzhou University, China, 2012) and the Standards for the Administration of Experimental Practices (Jiangsu, China, 2008). We also confirm that the field studies did not involve endangered or protected species.

### 2.2. Experimental Animals

A total of 180 fertilized eggs of Rugao chickens were collected from the Poultry Institute, Chinese Academy of Agricultural Sciences. In order to isolate ESCs in vitro, 60 newly laid fertilized eggs were randomly selected, and the remaining eggs were incubated in an incubator at 37.8 °C and 60% relative humidity. After incubation for 4.5 days, 60 fertilized eggs were taken to isolate PGCs, and the remaining 60 eggs were incubated for 18.5 days for SSC isolation.

### 2.3. Cell Isolation and Flow Cytometry

The specific separation and cultivation methods of chicken ESCs, PGCs and SSCs were performed as reported by Zuo et al. [18] Before cell isolation, all eggs were disinfected with bromogeramine and 75% anhydrous ethanol. To isolate ESCs, blastoderms were separated from newly laid fertilized eggs and digested with trypsin-EDTA (0.25%) for 3 min. Then, cells were filtered and differentially cultured for 24 h. The cells were separated from the supernatant for flow cytometry.

To isolate PGCs, embryos were isolated from eggs which had been incubated for 4.5 days. Genital ridges from embryos were collected and cut up to isolate cells. Then, trypsin-EDTA (0.25%) was used to digest cells for 5 min. After differential attachment for 1 h, cells were separated from the supernatant for flow cytometry.

To isolate SSCs, male embryos from eggs incubated for 18.5 days were collected. Testes were isolated and cut up completely to isolate cells. Collagenase I and trypsin-EDTA (0.25%) were used to digest cells. After differential attachment for 45 min, cells were collected for flow cytometry.

After cell isolation, FACSAria SORP (BD Company, Franklin Lakes, NJ, USA) was applied to obtain highly purified ESCs, PGCs and SSCs labeled by two antibodies. Antibodies to Nanog and Oct4 (Abcam, Cambridge, UK, dilution ratio 1:100) were used to identify ESCs. while antibodies to SSEA (BioLengend, San Diego, CA, USA, dilution ratio 1:100) and C-kit (Southern Biotech, Birmingham, AL, USA, dilution ratio 1:100), and integrin α6 and integrin β1 (BioLengend, San Diego, CA, USA, dilution ratio 1:100) were used to identify PGCs and SSCs, respectively.

### 2.4. Library Construction and Sequencing

Total RNA of cells isolated by flow cytometry was extracted. RNA purity and concentration were checked by a NanoDrop. RNA degradation and contamination were analyzed by agarose gels and the integrity of RNA was detected accurately by an Agilent2100. After RNA detection was done, ribosomal RNA was removed using the Epicenter Ribo-ZeroTM rRNA Removal Kit. Then, 3 μg total RNA of each cell sample were fragmented into short fragments of 250–300 bp by adding fragmentation buffer. Taking these short RNA fragments as a template, the first strand cDNA was synthesized using random hexamers, and the second strand cDNA was subsequently synthesized by adding buffer, dNTPs (dUTP, dATP, dGTP and dCTP) and DNA polymerase I. AMPure XP beads were used to purify double-stranded cDNA. After that, double-stranded cDNA was then repaired, poly A was added and sequencing adapters were connected. Then, AMPure XP beads were used to select the cDNA fragments (150-200), and USER (Uracil-Specific Excision Reagent) enzyme was used to degrade the second strand cDNA containing uracil. Lastly, the chain-specific cDNA library was constructed by PCR enrichment.

### 2.5. Sequence Data Analysis

In order to ensure the accuracy of subsequent analysis, raw reads were filtered by removing low-quality reads with adapters and a ratio of poly-N greater than 10%. Then, filtered data were mapped to the *Gallus* genome using Tophat2 [19]. Based on the transcription splicing results using cufflinks [20], transcripts with unknown chain direction were removed, while those of length ≥200 bp and an exon number ≥2 were retained. Then, candidate transcripts of different types (long intergenic non-coding RNAs (lincRNA), intronic lncRNA and anti-sense lncRNA) were screened according to class code (http://cole-trapnell-lab.github.io/cufflinks/cuffcompare/index.html#transfrag-class-codes). After that, phyloCSF [21], CPC [22] and PFAM [23] were used to predict the coding potential of lncRNAs, and the intersection of the three screening methods was selected as candidate lncRNAs for subsequent analysis.

### 2.6. Prediction and Functional Enrichment Analysis of lncRNA Target Genes

According to the prediction of lncRNA functions originating from the functional annotations of their related target mRNAs, protein-coding genes 100 kb upstream and downstream of lncRNAs were defined as potential cis-regulated target genes. Functional enrichment analysis of potential cis-regulated target genes was carried out by GOseq [24] based on Wallenius non-central hyper-geometric distribution and Kyoto Encyclopedia of Genes and Genomes (KEGG), with which the most important biochemical metabolic pathways and signal transduction pathways in which specific genes are involved can be identified. The value of *p* < 0.05 in this analysis denotes significant enrichment of GO terms and KEGG pathways.

### 2.7. The lncRNA–mRNA and mRNA–mRNA Co-expression Network Analysis

The lncRNAs and mRNAs could be correlated through their target relationship to construct an lncRNA–mRNA interaction network and, meanwhile, combining the mRNA–mRNA protein co-expression network of lncRNA cis-target genes could further predict the potential biological functions of lncRNAs. According to the results of GO and KEGG pathway functional enrichment analysis, differentially expressed lncRNAs (|log2FC > 6|) associated with germ cell development were selected for further analysis. Correlations of lncRNA–mRNA and mRNA–mRNA between differentially expressed lncRNAs and cis-target genes were analyzed by STRING (http://string-db.org/), and Cytoscape software [25] was used to construct the final co-expression network.

### 2.8. Quantitative Real-Time Polymerase Chain Reaction (qRT-PCR) Verification

The RNAs from ESCs, PGCs and SSCs used for RNA-seq were simultaneously used for qRT-PCR analysis. A total of 1 μg RNA of each cell sample was converted into cDNA using FastKing gDNA Dispelling RT SuperMix (TIANGEN, Beijing, China) according to the manufacturer’s instructions. Five differentially expressed lncRNAs were randomly selected to verify the accuracy of RNA-seq using qRT-PCR, and β-actin was used as a reference gene (Table 1). The accession numbers and sequences of lncRNAs are shown in Appendix A. All primers were designed by Primer Premier 5.0 software (Premier Biosoft, San Francisco, CA, USA).

SuperReal PreMix Color (SYBR Green) (TIANGEN BIOTECH, Beijing, China) was used for qRT-PCR amplification. Each group had three replicates. The PCR amplification procedure was set as follows: initial denaturation at 95 °C for 15 min and 40 cycles for denaturation (95 °C, 10 s) and annealing (60 °C, 30 s). Moreover, qRT-PCR was performed using the CFX Connect Real-Time System (BIO-RAD, CA USA).

The 2^-ΔΔCt^ method was used to calculate the relative gene expression [26]. Statistical analysis was performed with SPSS 17.0 software (SPSS, Inc., Chicago, IL, USA). All data are presented as mean ± SEM. The levels of gene expression were analyzed for significant differences with one-way ANOVA. A probability of *p* < 0.05 was considered to be statistically significant.

## 3. Results

### 3.1. Overview of Sequencing Results

Before the RNA-seq, chicken ESCs, PGCs and SSCs were collected by flow cytometry after double labeling of specific antibodies (Figure 1). Total RNAs of sorted cells were extracted and a cDNA library was constructed by Illumina 2500. A total of 703,283,306 raw reads were gathered, in which the proportion of N (unable to determine base information) was more than 10% and the adapters were removed. As result, the number and proportion of clean reads in ESCs, PGCs and SSCs were 126,200,470 (64.46%), 142,487,396 (83.75%) and 106,333,040 (74.24%) (Table 2). Coding potential analysis methods (CPC, PFAM and phyloCSF) were used to identify the non-coding transcripts in ESCs, PGCs and SSCs. The intersection results of the three software analyses showed that 10,281 non-coding transcripts were obtained, which were used as the lncRNA data sets for subsequent analysis (Figure 2A).

Based on a preliminary analysis of the structure, about 59% of the lncRNAs had a length of between 200 bp and 1000 bp, while about 1% of the lncRNAs had a length of more than 10,000 bp (Figure 2B). Most of the lncRNAs had a length of the Open Reading Frame (ORF) between 50 bp and 250 bp, about 3% of lncRNAs had a length of the ORF of less than 50 bp and about 2% of lncRNAs had a length of the ORF of more than 250 bp (Figure 2C). Moreover, 86% of lncRNAs had two to three exons, while lncRNAs with more than eight exons accounted for only 1% of the total (Figure 2D).

### 3.2. Analysis and Verification of Differentially Expressed lncRNAs in ESCs, PGCs and SSCs

To identify the characteristics of lncRNA expression in ESCs, PGCs and SSCs, the expression levels of lncRNAs in three kinds of cells were estimated by fragments per kilobase per million (FPKM) [20]. Hierarchical clustering and heatmaps of lncRNAs showed that the three kinds of cells clustered separately from each other (Figure 3A). Differentially expressed lncRNAs (DELs) (|log2(fold change)| ≥ 2 (*p* < 0.05)) were screened out, and 205 up-regulated and 140 down-regulated DELs were found in ESCs vs. SSCs, 217 up-regulated and 140 down-regulated DELs were found in PGCs vs. SSCs and 195 up-regulated and 79 down-regulated DELs were found in ESCs vs. SSCs. DELs in each group were further subdivided according to the fold change. The vast majority of DELs were in the range of 4 ≤ |log2FC| < 6, only a few lncRNAs were distributed in the range of |log2FC| ≥ 10 (Figure 3B–D). Moreover, the number of up-regulated (blue) DELs in three groups was larger than those that were down-regulated (red). In particular, the number of up-regulated or down-regulated DELs in PGCs vs. SSCs was larger than that in the other two groups, suggesting that lncRNAs play an important role in the differentiation of PGCs to SSCs.

The accuracy of the differentially expressed lncRNAs identified by RNA-seq was validated by qRT-PCR. Five lncRNAs, which were significantly differentially expressed in ESCs vs. PGCs, ESCs vs. SSCs and PGCs vs. SSCs, were randomly selected for qRT-PCR verification. The result showed that the qRT-PCR data confirmed the RNA-seq results, indicating that the RNA-seq data were reliable and could be used for the following analysis (Figure 3E).

### 3.3. Prediction and Functional Enrichment Analysis of lncRNA Target Genes

In order to identify the potential function of DELs in ESCs vs. PGCs, ESCs vs. SSCs and PGCs vs. SSCs, cis-regulated target genes of DELs were analyzed by GO analysis and KEGG pathway analysis. GO functional analysis showed that DELs in ESCs vs. PGCs were significantly enriched in biological processes such as reaction to reactive oxygen species (GO:0000302), positive regulation of cellular biosynthesis process (GO:0031328), positive regulation of stem cell differentiation (GO:2000738) and germ-line stem cell division (GO:0042078) (Figure 4A, Appendix A). KEGG enrichment analysis showed that autophagy (ko04140), steroid hormone biosynthesis (ko00140), caffeine metabolism (ko00232) and TGF-β signaling pathway (ko04350) were significantly enriched (Figure 4B, Appendix A). In ESCs vs. SSCs, DELs were significantly enriched in biological processes such as positive regulation of stem cell differentiation (GO:2000738), germ-line stem cell division (GO:0042078) and spermatogenesis (GO:0007283) (Figure 4C, Appendix A). KEGG enrichment analysis showed that autophagy (ko04140), Wnt/β-catenin signaling pathway (ko04310), Notch signaling pathway (ko04330) and ErbB signaling pathway (ko04012) (Figure 4D, Appendix A) were significantly enriched. In PGCs vs. SSCs, DELs were significantly enriched in biological processes such as Sertoli cell proliferation (GO:0060011), Sertoli cell proliferation (GO:0060011) and cell motility (GO:0048870) (Figure 4E, Appendix A). KEGG enrichment analysis showed that autophagy (ko04140), ErbB signaling pathway (ko04012) and focal adhesion (ko04510) were enriched (Figure 4F, Appendix A). According to the above functional enrichment analysis, DELs in the three cell groups were significantly enriched in autophagy and the ErbB signaling pathway, indicating that these two signaling pathways may play important roles in the differentiation of chicken ESCs into SSCs.

### 3.4. Screening of DELs Related to Germ Cell Development

To identify the DELs related to germ cell development in the differentiation of ESCs into SSCs, DELs in ESCs vs. PGCs, ESCs vs. SSCs and PGCs vs. SSCs were further screened and combined with the results of GO and KEGG pathway analysis. DELs which were enriched in the biological function of germ cell development in the three groups were selected. As shown in Appendix A, a total of 22 DELs (17 up-regulated, five down-regulated) were selected in ESCs vs. PGCs and 42 DELs (35 up-regulated, seven down-regulated) were selected in ESCs vs. SSCs (Appendix A). However, 11 DELs were found in PGCs vs. SSCs. Interestingly, these 11 DELs were all down-regulated, that is, the expression level of DELs in ESCs was significantly higher than that in PGCs (Appendix A), suggesting these DELs may play a pivotal role in the formation and differentiation of SSCs.

### 3.5. Co-expression Network Analysis of lncRNA–mRNA and mRNA–mRNA

For further identifying the potential regulation of DELs in the differentiation of ESCs into SSCs, DELs (|log2FC| > 6) were selected as candidate lncRNAs from the above screened DELs related to germ cell development (Appendix A). The protein interactions among target genes of candidate lncRNAs were analyzed online using the STRING database (https://string-db.org/). Combined with Cytoscape software [25], the interaction networks between lncRNAs and target genes (lncRNA–mRNA), and all target genes (mRNA–mRNA) in each cell group were constructed.

In ESCs vs. PGCs, a total of 13 DELs (11 up-regulated, two down-regulated) related to germ cell development were screened out (Appendix A), and 91 cis-target genes of these lncRNAs were predicted, such as CYP1A1 [27], STRA8 [28], WDR91 [29], SOX9 [30] and BMP7 [31], which were all reported to be involved in germ cell differentiation or proliferation, which further indicated that lncRNAs (XLOC_160716, XLOC_023952, XLOC_240662, XLOC_362463) corresponding to these five target genes play roles in the development of germ cells (Figure 5A). Interestingly, we found that an up-regulated (regular triangle) lncRNA (XLOC_612026) and a down-regulated (inverted triangle) lncRNA (XLOC_612029) have the same target gene (BMPR-Ⅱ) (Figure 5A). Additionally, the mRNA–mRNA co-expression network showed that SUMO1 and NOP58, which were cis-target genes of XLOC_612026, had the highest connectivity among other target genes (Figure 5B), indicating a significant co-expression relationship between SUMO1 and NOP58. Notably, there was another co-expression relationship between NOP58 and the target gene (WDR12) of XLOC_612029 (Figure 5B). Therefore, it can be concluded that the opposite expression pattern of XLOC_612026 and XLOC_612029 in ESCs vs. PGCs may be related to the interaction between their target genes. In addition, an up-regulated lncRNA (XLOC_674549) which had 12 cis-target genes showed that four of its target genes (APTX, SUM1, XPA, DNAJA1) had a co-expression relationship (Figure 5A,B), illustrating that the regulation of XLOC_674549 in the differentiation of ESCs into PGCs may be affected by the interaction of its target genes.

In ESCs vs. SSCs, a total of 21 DELs (18 up-regulated, three down-regulated) were screened out from the 42 DELs (Appendix A), of which 11 DELs were consistent with those that were screened in ESCs vs. PGCs, and 110 cis target genes were predicted. Among the 11 DELs, XLOC_023952, XLOC_240662 and XLOC_362463 were also predicted to be involved in regulating the differentiation of PGCs to SSCs due to the function of their target genes. Moreover, as in the above analysis in ESCs vs. PGCs, XLOC_612026 and XLOC_612029 were also predicted to regulate the differentiation of ESCs to SSCs, that is, both XLOC_612026 and XLOC_612029 play regulatory roles in male germ cell formation. Notably, XLOC_612029 and ALDBGALG0000001810 were expressed more highly in PGCs and SSCs than in ESCs (Figure 6A, Appendix A), suggesting that these two lncRNAs may play a positive role in the process of male germ cell formation. The mRNA–mRNA co-expression network showed that the target gene (EIF2S3) of XLOC_022100 had a strong connection with the neighboring gene DRG1, one target gene of ALDBGALG0000001810, indicating that XLOC_022100 was involved in male germ cell formation.

According to the screening criteria (|log2FC| > 6), only three DELs (ALDBGALG0000002986, XLOC_161226, XLOC_657380) were screened from the 12 DELs related to germ cell development in PGCs vs. SSCs. All these DELs were down-regulated, which means these lncRNAs were highly expressed in SSCs (Figure 7A, Appendix A). Here, mTOR, one important regulator in cell development and proliferation, was predicted to be a target gene of ALDBGALG0000002986 (Figure 7A). A previous study revealed that the mTOR signaling pathway is essential to regulate cell renewal, proliferation and differentiation of spermatogonial stem cells [32,33,34]. Therefore, ALDBGALG0000002986 may play the same role in the differentiation of PGCs to SSCs through the mTOR signaling pathway. The mRNA–mRNA co-expression network showed that target genes FXR1, TCF4 and DNAJC19 of a down-regulated lncRNA (XLOC_6573809) were found to have strong co-expression relationships with each other. Therefore, XLOC_6573809 was predicated to play an important role in the differentiation process from PGCs to SSCs through its target genes.

## 4. Discussion

Recently, emerging evidence revealed that a large number of lncRNAs are widely involved in regulating the development of the reproduction system [35]. However, there is still a lack of studies on the role of lncRNAs in the formation of male germ cells. The chicken is a classic model in development biology [36]. Based on this model, studying the function of lncRNAs in the development of male germ cells is helpful to further understand the development process of germ cells. In this study, high-throughput transcriptome sequencing was used to identify the expression profile of lncRNAs in chicken ESCs, PGCs and SSCs, and bioinformatics was used to predict the potential function of lncRNAs. As a result, several specially expressed lncRNAs were found to be involved in regulating the differentiation of ESCs to SSCs through a functional annotation analysis.

A large number of studies have demonstrated that lncRNAs regulate the expression of their target protein-coding genes via transcriptional co-activation or co-repression [37,38]. Here, target genes of DELs in ESCs vs. PGCs, ESCs vs. SSCs and PGCs vs. SSCs were predicted and analyzed, such as CYP1A1, Stra8, SOX9, WDR91 and BMP7. In this study, CYP1A1 was predicted to be the target gene of XLOC_160716 in ESCs vs. PGCs. Li et al. [27] reported that CYP1A1 can regulate the differentiation of male chicken germ cells through cytochrome P450 based on exogenous metabolism. Hence, we speculated that XLOC_160716 participated in the differentiation of ESCs to PGCs by targeting CYP1A1. Similarly, in ESCs vs. PGCs and ESCs vs. SSCs, Stra8 [39] and WDR91 [29] were target genes of XLOC_023952, as both of them had been proved to be involved in the development of germ cells. Therefore, XLOC_023952 was likely to regulate the differentiation from ESCs to SSCs through its target genes (STRA8, WDR91). Sox9 is a member of the Sox gene family. Matsumoto et al. [30] found that knockout or overexpression of Sox9 caused sexual reversal in mouse embryos, indicating that Sox9 plays an important role in sex determination. In this study, Sox9 was predicted to be a target gene of XLOC_240662 both in ESCs vs. PGCs and ESCs vs. SSCs. As a result, XLOC_240662 could be speculated to participate in germ cell differentiation, especially in sex determination. BMP7, one target gene of XLOC_362463 both in ESCs vs. PGCs and ESCs vs. SSCs, was reported to regulate germ cell proliferation in mice [31], which indicated that XLOC_362463 may be a regulator in germ cell development. To sum up, the function of these lncRNAs above was predicted regarding their target genes’ function. However, whether they are involved in germ cell development still needs to be further verified by experiments.

The functional annotation analysis showed that DELs in ESCs vs. PGCs, ESC vs. SSCs and PGCs vs. SSCs were mainly enriched in autophagy, the TGF-β signaling pathway, the Wnt/β-catenin signaling pathway, the ErbB signaling pathway, Notch signaling pathways, etc. To date, many pathways have been revealed to be involved in male germ cell differentiation, such as Wnt [40], TGF-β/BMP [41], Notch [42], MAPK [43], JAK-STAT [44] signaling pathways, etc. In this study, BMP7 was predicted to be the target gene of XLOC_362463 in ESCs vs. PGCs and ESCs vs. SSCs. As one of the members in the TGF-β superfamily, BMP7 was required for PGC, spermatogonia stem cell and Sertoli cell differentiation or proliferation in mice [45,46]. This information provided evidence for XLOC_362463 regulating the formation of SSCs through the TGF-β/BMP signaling pathway. In ESCs vs. SSCs, GO analysis showed that DELs related to germ cell development were significantly enriched in the GO term of the negative regulation of the canonical Wnt signaling pathway. The Wnt signaling pathway has been reported to be involved in female germ cell development and the differentiation of chicken ESCs to SSCs [47,48]. Coincidently, we found that the target gene SOX9 of XLOC_240662 was enriched in this GO term. Therefore, we hypothesized that XLOC_240662 may regulate the differentiation from ESCs to SSCs by targeting SOX9.

Notably, according to the functional annotation analysis, the autophagy signaling pathway was significantly enriched in ESCs vs. PGCs, ESCs vs. SSCs and PGCs vs. SSCs. Pannerdoss et al. [49] found that cross-talk between miR-471-5p and autophagy component proteins could regulate LC3-associated phagocytosis (LAP) to clear apoptotic germ cells. However, whether lncRNAs regulate autophagy in male germ cell formation remains unclear. In this study, mTOR was predicted to be a target gene of ALDBGALG0000002986 in PGCs vs. SSCs. Previous studies have showed that mTOR was involved in cell renewal, proliferation, differentiation and meiosis of spermatogonial stem cells [33,34], suggesting that ALDBGALG0000002986 may regulate the differentiation process from PGCs to SSCs. Recently, mTOR has been identified as an essential factor to mediate autophagy with p53 to regulate the fate of spermatogonial progenitor cells (SPCs) [50]. Hence, ALDBGALG0000002986 is likely to be involved in autophagy in the differentiation process from PGCs to SSCs through targeting mTOR.

According to the interaction network, the target gene SUMO1 of XLOC_616026 was shown to be the key gene in the network modules of ESCs vs. PGCs and ESCs vs. SSCs. Emerging evidence showed that SUMO is involved in regulating cell proliferation, differentiation, cell cycle, apoptosis and cell stress [51]. Cossec et al. [52] revealed that inhibiting SUMO could promote the transformation of ESCs into totipotent 2C-like cells, indicating that SUMO is a key factor in stabilizing cellular pluripotency. Moreover, increasing evidence shows that SUMO is involved in autophagy, for example, the overexpression of the SUMO1 or SUMO2/3 enzyme (SENP3) accelerated the accumulation of autophagosomes [53,54]. Thus, in addition to mTOR, SUMO1 provided further evidence that autophagy was involved in germ cell differentiation from ESCs to SSCs. Vigodner et al. [55] found that human testicular SUMO1 is involved in both facultative and constitutive heterochromatin processes in spermatocytes. This provides a possibility for SUMO1 to participate in the differentiation process of chicken ESCs to SSCs. Interestingly, the interaction network also showed that SUMO1 was co-expressed with NOP58, which was predicted to be another target gene of XLOC_616026. NOP58 has been identified to be a substrate for SUMOylation in regulating protein function [56] and, meanwhile, SUMOylation was widely reported to regulate meiosis in mammalian oocytes and male germ cells [57]. Although no studies have been clearly reported that lncRNAs are involved in germ cell differentiation through SUMOylation, our results may provide some evidence for this aspect.

## 5. Conclusions

In summary, 296, 280 and 357 differentially expressed lncRNAs were screened in ESCs vs. PGCs, ESCs vs. SSCs and PGCs vs. SSCs, respectively. Functional annotation analysis showed that autophagy, TGF-β, Wnt/β-catenin, ErbB and Notch signaling pathways were involved in male germ cell formation. By analyzing interaction networks, we also revealed that several key lncRNAs, such as XLOC_612026, XLOC_612029, XLOC_240662, XLOC_362463, XLOC_023952, XLOC_674549, XLOC_160716, ALDBGALG0000001810, ALDBGALG0000002986, XLOC_674549, XLOC_022100 and XLOC_657380, are involved in male germ cell formation. As a whole, this study can not only provide data support for constructing gene networks of male germ cell formation but also provide new ideas for improving the induction efficiency of germ cells in vitro.

## Figures and Tables

**Figure 1 animals-10-01850-f001:**
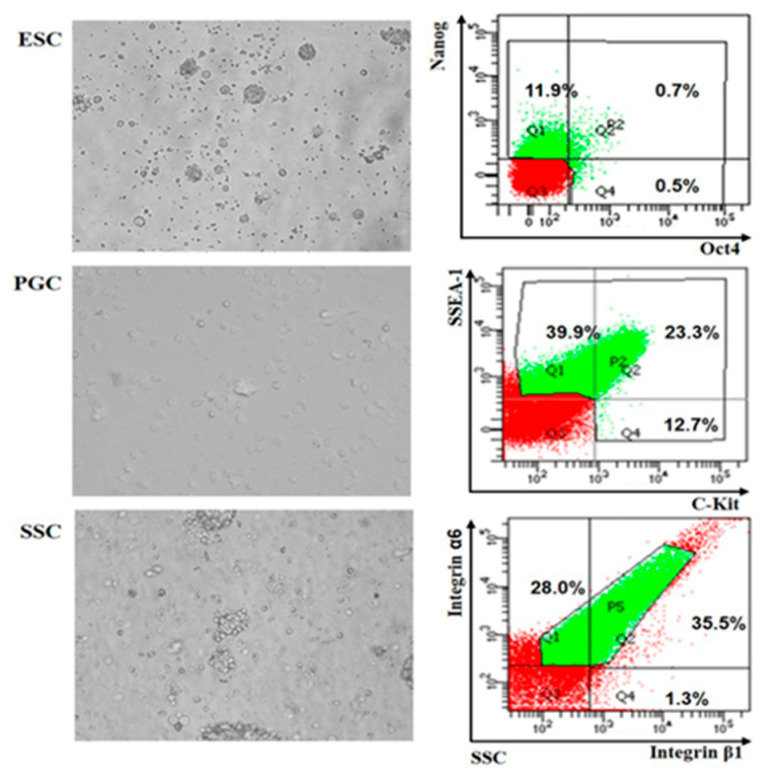
Embryonic stem cell (ESC), primordial germ cell (PGC) and spermatogonial stem cell (SSC) isolation and identification. Chicken ESCs, PGCs and SSCs were isolated (400×), identified with Nanog and Oct4, SSEA-1 and C-Kit and integrin α6 and integrin β1 by flow cytometry, respectively.

**Figure 2 animals-10-01850-f002:**
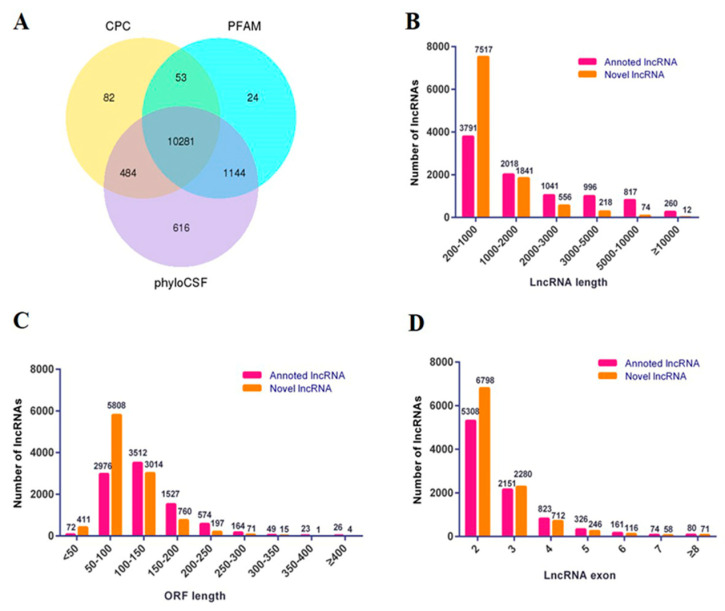
Summary of the predicted lncRNAs. (**A**) Filtering of candidate non-coding RNAs. Venn diagram of coding potential analysis using Coding Potential Calculator (CPC), Pfam-scan (PFAM) and phylogenetic codon substitution frequency (phyloCSF). Those shared by these three methods were predicted as candidate non-coding RNAs. (**B**) Classification of the length of annotated lncRNAs and novel lncRNAs. (**C**) Classification of the open reading frame (ORF) length of annotated lncRNAs and novel lncRNAs. (**D**) Classification of the exon number of annotated lncRNAs and novel lncRNAs.

**Figure 3 animals-10-01850-f003:**
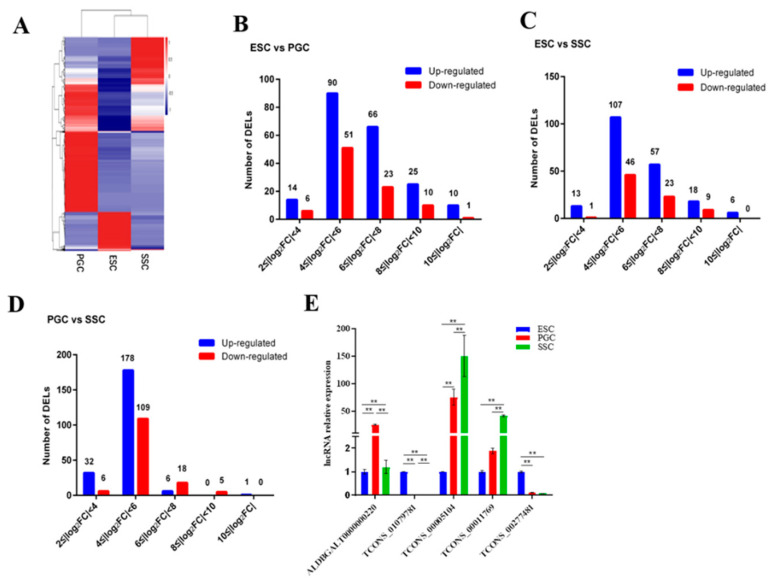
Overview of the expression profile of differentially expressed lncRNAs (DELs) in chicken ESCs, PGCs and SSCs. (**A**) Heatmaps of differential expression of lncRNAs (DELs) in chicken ESCs, PGCs and SSCs. (**B**–**D**) The absolute value of the fold change was divided into five groups. Each group shows the number of up-regulated (blue) and down-regulated (red) DELs. (**B**) DELs in ESCs vs. PGCs, (**C**) DELs in ESCs vs. SSCs, (**D**) DELs in PGCs vs. SSCs. (**E**) The qRT-PCR validation of five differentially expressed lncRNAs (DELs) selected from the RNA-seq transcriptome (* represents *p* < 0.05 and ** represents *p* < 0.01).

**Figure 4 animals-10-01850-f004:**
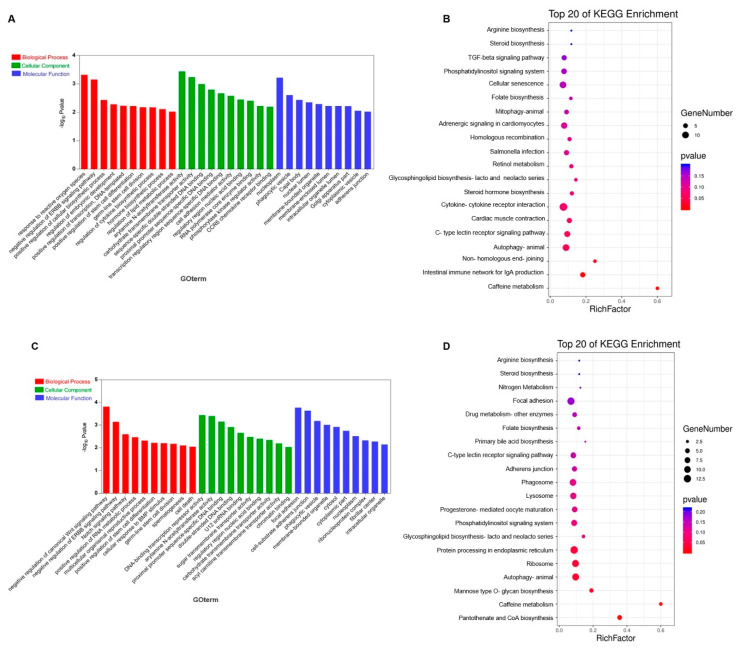
Gene ontology (GO) and Kyoto Encyclopedia of Genes and Genomes (KEGG) analysis of differentially expressed lncRNAs in ESCs vs. PGCs, ESCs vs. SSCs and PGCs vs. SSCs. (**A**,**B**) The GO enrichment terms and pathways in ESCs vs. PGCs. (**C**,**D**) The GO enrichment terms and pathways in ESCs vs. SSCs. (**E**,**F**) The GO enrichment terms and pathways in PGCs vs. SSCs.

**Figure 5 animals-10-01850-f005:**
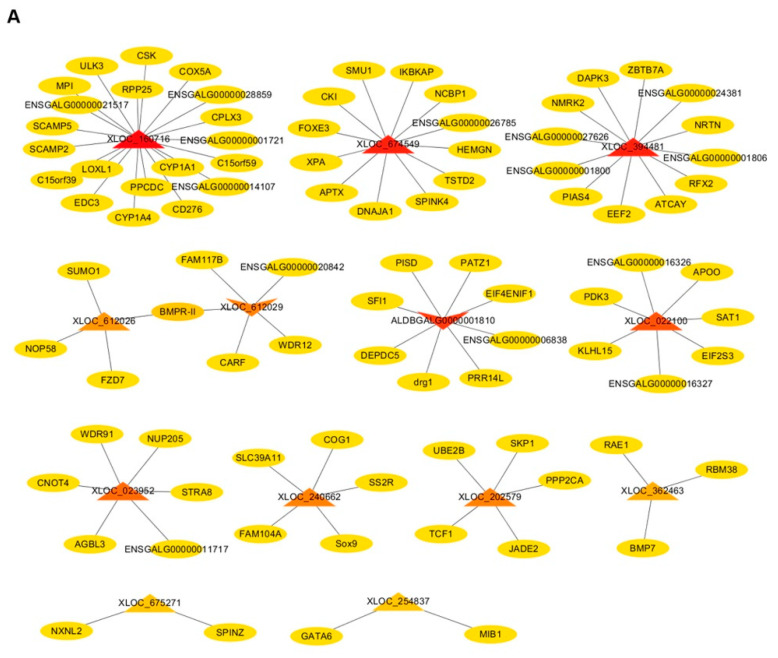
The interaction network of lncRNA–mRNA, mRNA–mRNA in ESCs vs. PGCs. (**A**) The interaction relationship between candidate differentially expressed lncRNAs (DELs) and target genes (lncRNA–mRNA) in ESCs vs. PGCs. Circular nodes and triangular nodes represent target genes and DELs, respectively. Regular triangles and inverted triangles represent up-regulated and down-regulated DELs, respectively. The depth of the triangle color indicates the number of target genes. The darker the triangle color is, the more target genes the DEL has. (**B**) The interaction relationship of target genes of candidate DELs. Circular nodes represent target genes of DELs. The depth of the color indicates the weight of the target gene in the network. The darker the color is, the higher the weight of the target gene, indicating the target gene has stronger connections with neighboring genes.

**Figure 6 animals-10-01850-f006:**
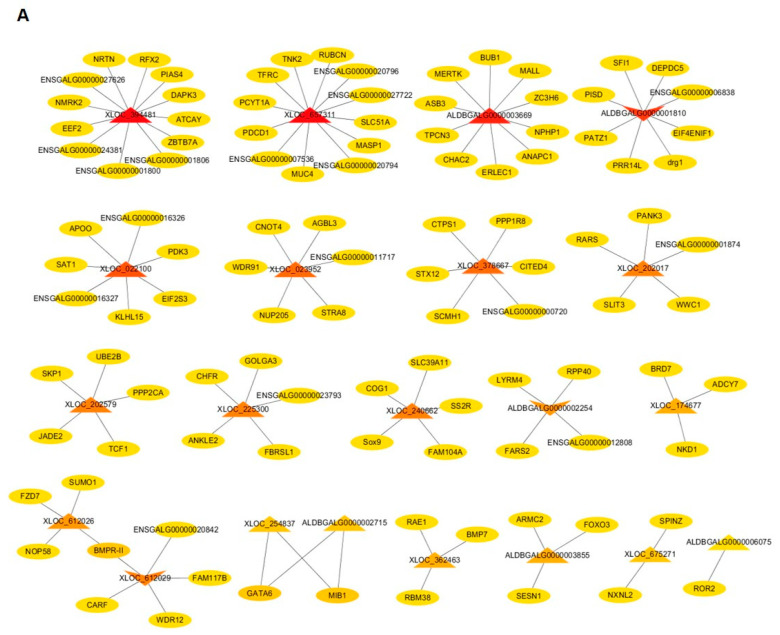
The interaction network of lncRNA–mRNA, mRNA–mRNA in ESCs vs. SSCs. (**A**) The interaction relationship between candidate differentially expressed lncRNAs (DELs) and target genes (lncRNA–mRNA) in ESCs vs. SSCs. Circular nodes and triangular nodes represent target genes and DELs, respectively. Regular triangles and inverted triangles represent up-regulated and down-regulated DELs, respectively. The depth of the triangle color indicates the number of target genes. The darker the triangle color is, the more target genes the DEL has. (**B**) The interaction relationship of target genes of candidate DELs. Circular nodes represent target genes of DELs. The depth of the color indicates the weight of the target gene in the network. The darker the color is, the higher the weight of the target gene, indicating the target gene has stronger connections with neighboring genes.

**Figure 7 animals-10-01850-f007:**
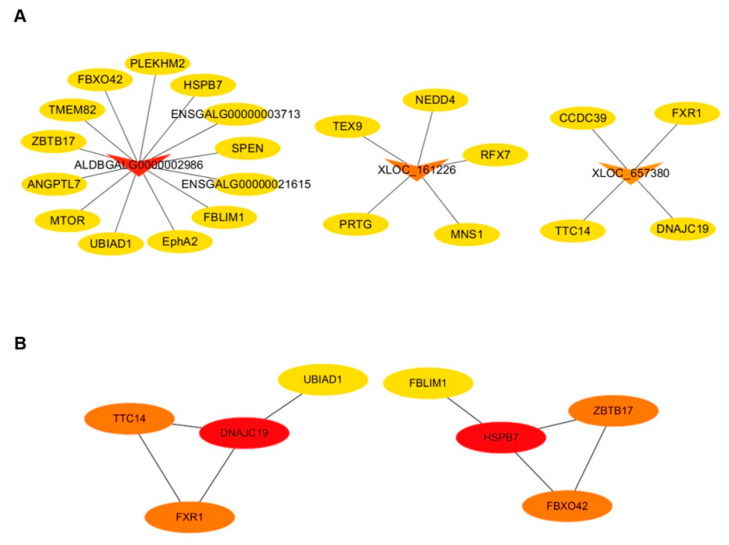
The interaction network of lncRNA–mRNA, mRNA–mRNA in PGCs vs. SSCs. (**A**) The interaction relationship between candidate differentially expressed lncRNAs (DELs) and target genes (lncRNA–mRNA) in PGCs vs. SSCs. Circular nodes and triangular nodes represent target genes and DELs, respectively. Regular triangles and inverted triangles represent up-regulated and down-regulated DELs, respectively. The depth of the triangle color indicates the number of target genes. The darker the triangle color is, the more target genes the DEL has. (**B**) The interaction relationship of target genes of candidate DELs. Circular nodes represent target genes of DELs. The depth of the color indicates the weight of the target gene in the network. The darker the color is, the higher the weight of the target gene, indicating the target gene has stronger connections with neighboring genes.

**Table 1 animals-10-01850-t001:** Primers for qRT-PCR.

Long Non-Coding RNA/mRNA	Forward Primers (5′–3′)	Reverse Primers (5′–3′)	Product Size (bp)
ALDBGALT0000000220	GTTAAGTGCAGCAAGACT	GAAGATGAGAACCAGAAA	110
TCONS_01079781	AAGTACTAGAAGCCAGCT	GTTCTGTCTTAGCGATGT	81
TCONS_00005104	AGAAGTCAGTGAAGAGCAGGAA	TCAAGCCCAGAACCCAAA	134
TCONS_00011769	AAACGGATAATGACAAGA	ATTGCTTTCCCTGTATTT	115
TCONS_00277481 β-actin	AATAAATACTTCGGGTGA CAGCCATCTTTCTTGGGTAT	CTTTGACGTACTTTGTGC CTGTGATCTCCTTCTGCATCC	150 169

**Table 2 animals-10-01850-t002:** Summary of whole-transcriptome sequencing.

Sample Name	Raw Reads	Clean Reads	Clean Bases	Error Rate (%)	Q20 (%)	Q30 (%)	GC Content (%)
ESC	195781306	126200470	18.93G	0.03	96.60	92.38	48.85
PGC	170137682	142487396	21.27G	0.02	96.95	93.02	47.83
SSC	143227516	106333040	15.80G	0.03	95.63	90.78	48.62

## Data Availability

The sequencing data of this paper has been submitted to SRA with the Accession No. SRR12145635, SRR12145634 and SRR12145633.

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
