# Peer review of "Analysis of lncRNA Expression Profile during the Formation of Male Germ Cells in Chickens"

_animals, 2020, doi:10.3390/ani10101850_

Round 1

Reviewer 1 Report

Responses to the reviewer’s comments are unacceptable that this reviewer cannot find in the revision.

Point 1: This reviewer strongly suggests that introduction needs description for "LncRNA cross-talk with epigenetic factors and association with epigenetic modification and transcriptional regulation" to justify the current study as authors mentioned in lines 27 and 334.

Response 1: Thank you for your suggestion. As you suggested, it’s necessary to add description of “lncRNA cross-talk with epigenetic factors”. We have made a major revision to the discussion section in the revised manuscript.

Please show and indicate specifically what authors did in the discussion section with the comprehension of reviewer’s comments.

Author Response

Response to Reviewer 1 Comments

Point 1: This reviewer strongly suggests that introduction needs description for "LncRNA cross-talk with epigenetic factors and association with epigenetic modification and transcriptional regulation" to justify the current study as authors mentioned in lines 27 and 334.

Response 1: Dear reviewer, thank you very much for your suggestion. I’m deeply sorry for that we didn’t understand your suggestion in our first revision. In this revision, we added the description of the cross-talk between lncRNAs and other epigenetic factors to justify the current study mentioned in lines 27 and 334.

In the introduction part, we added the mechanism of action of lncRNAs regulate gene expression at three levels, which are epigenetic, transcriptional and post-transcriptional levels. For instance, lncRNAs can interface with chromatin remodeling machinery in several ways, including acting as signal lncRNAs or scaffold lncRNAs by recruiting chromatin remodeling complexes to specific gene locus. What’s more, lncRNAs can work as ceRNAs to modulate transcription by sequestering transcription factors, catalytic proteins or miRNAs. Additionally, we added several references about how lncRNAs regulate the development of germ cells in spermatogenesis association with epigenetic modification and transcriptional regulation (marked with yellow). In the discussion part, we added the description that lncRNAs can regulate the expression of their target protein-coding genes via transcriptional co-activation/repression to justify that the lncRNAs we selected may involve in the differentiation of chicken ESC to SSC via association with their target genes.

Reviewer 2 Report

Missing citation for some softwares in the method.

I did not see anything about qPCR primer efficiency. 

Author Response

Thank you for your suggestion

Round 2

Reviewer 1 Report

Comments: Line 77: please change LncRNA to lncRNA. Line 429: please change mTRO to mTOR.

Author Response

This manuscript is a resubmission of an earlier submission. The following is a list of the peer review reports and author responses from that submission.

Round 1

Reviewer 1 Report

Authors investigated the key lncRNAs involved in the chicken male germ cell formation by RNA-seq analysis.

Major comments:

This reviewer strongly suggests that introduction needs description for "LncRNA cross-talk with epigenetic factors and association with epigenetic modification and transcriptional regulation" to justify the current study as authors mentioned in lines 27 and 334.

Minor comments:

  1. Lines 16-17, and 29: please change to "spermatogonial stem cell (SSC)".
  2. Lines 382, 383, and 397: please change “spermatogonial stem cells” to “SSCs”.
  3. Line 69: please change “LncPGCAT-1” to “LncRNA PGC transcript-1”.
  4. Line 125: please change “sNTPs” to “dNTPs”.
  5. Line 140: please change “IncRNAS” to “IncRNAs”.
  6. Line 166: please change “2-ΔΔCt” to “2-ΔΔCt ”.
  7. Line 238: please change “DLEs” to “DELs”.
  8. Line 353: please correct grammar in “base on”
  9. Line 361: please change “AL DBGALG0000002986” to “ALDBGALG0000002986”.
  10. Line 367: please change “spermatogonia stem cells” to “SSCs”
  11. Line 375: please change “North” to “Notch”.
  12. Line 389: please change “AT” to “At”.
  13. Line 400: please use “ubiquitin” rather than ubiquitination, even though it's not wrong.
  14. Line 413: please correct to “SUMO1 and NOP58”.
  15. Line 423: please change “studies” to “studied”.
  16. Line 425: please change “mouse primordial germ cells” to mPGCs

Reviewer 2 Report

Dear authors, 

I have suggested a revision of your manuscript before any acceptance for publications in animals to the editor. Here, i have listed my concerns/comments.

Introduction:

L45-56: Please be more clear and "less familiar". Maybe english should be revised in this section

L52: I suggest you "." instead of ",". So it would be a new sentence starting by: "The low efficiency ..."

L58: "little or no potential" ==> Are LncRNA translated into proteins sometimes ? I thought they were not. 

Methods: 

L103-112: Instead of listing the reagents used, please put them together with the steps where they have been employed. 

L114-116: Please describe a bit more the isolation protocol. 

Library construction and analysis: what is the quantity of RNA used? 

L159: "Cytoscape" instead of "cytoscope"? Citation for both softwares? 

RT-qPCR: How did you do the cDNA? With what? WHat is the PCR Cycle more less? Did you calculate the qPCR efficiency? Also, maybe link table 1 to this section and add efficiency if calculated. 

Results:

Figure 2a: Please add in the legend what are CPC, PFAM, PhyloCSF. 

Why don t you used software such as DESEQ2 instead of FPKM to calculate DE LncRNA?

Figure 3: What is Fig3e? missing in the legend. Is it possible to be more clear on the foldchange? Various ... is really not informative. 

Is it possible to be a bit more clear on the processes up and down regulated between cell types? Tables with the most represented or "interesting" GO for example. 

Fig4; Fig5, Fig6: Sorry it is really to small and quite hard to see anything. 

Table S2: Sorry i don t have access to the supplementary data. 

NEtworks: Please keep methods in methods and be more clear on the results by describing a bit more the selected lncRNA.Figure are very hard to follow and small. Can you clarify the used of the mRNA-mRNA network analysis?

Maybe insist more on the LncRNA that you think are active in germ cell proliferation/differentiation. 

Discussion: 

L334-L346: References? Look at the english sometimes.

The discussion must be improved. Indeed, it appears more as a listing of mechanisms implicated in stem cell biology rather than a real integration of your results. Also, for a non-specialist, it can be a bit confusing between what are you really bringing and what is known. So please be more clear. The conclusion must be developed. 
